# Superior EVOO Quality Production: An RGB Sorting Machine for Olive Classification

**DOI:** 10.3390/foods11182917

**Published:** 2022-09-19

**Authors:** Simona Violino, Lavinia Moscovini, Corrado Costa, Paolo Del Re, Lucia Giansante, Pietro Toscano, Francesco Tocci, Simone Vasta, Rossella Manganiello, Luciano Ortenzi, Federico Pallottino

**Affiliations:** 1Consiglio per la Ricerca in Agricoltura e l’Analisi dell’Economia Agraria (CREA)—Centro di Ricerca Ingegneria e Trasformazioni Agroalimentari, Via Della Pascolare 16, 00015 Monterotondo, Italy; 2Consiglio per la Ricerca in Agricoltura e l’Analisi dell’Economia Agraria (CREA), Centro di Ricerca Ingegneria e Trasformazioni Agroalimentari, Viale Lombardia C.da Bucceri, 65012 Cepagatti, Italy; 3Consiglio per la Ricerca in Agricoltura e l’Analisi dell’Economia Agraria (CREA), Centro di Ricerca Ingegneria e Trasformazioni Agroalimentari, Via Milano 43, 24047 Treviglio, Italy

**Keywords:** extra virgin olive oil, innovative classification, image analysis, quality, olive selection

## Abstract

Extra virgin olive oil (EVOO) is a commercial product of high quality, thanks to its nutritional and organoleptic characteristics. The olives ripeness and the choice of harvest time according to their color and size, strongly influences the quality of the EVOO. The physical sorting of olives with machines performing rapid and objective optical selection, impossible by hand, can improve the quality of the final product. The aim of this study concerns the classification of olives into two qualitative classes, based on the maturity stage and the presence of external defects, through an industrial RGB optical sorting prototype, evaluating its performance and comparing the results with those obtained visually by trained operators. EVOOs obtained from classified olives were characterized through chemical, physical-chemical analysis and sensory profile. For the first time, the optoelectronic technologies in an industrial system was tested on olives to produce superior quality EVOO. The selection allows late harvest, obtaining oils with good characteristics from fully ripe and unripe fruits together, separating defective olives with appropriate calibration and training. Optoelectronic selection creates the opportunity to blend the obtained oils destined to different applications according to the needs of the consumer or producer, using a vanguard technology at low cost.

## 1. Introduction

Extra virgin olive oil (EVOO) is one of the primary ingredients of the Mediterranean diet thanks to its organoleptic and health properties [1]. The 82% of the world production of EVOO comes from European Mediterranean countries. Indeed, the Mediterranean region, thanks to its dry and sub-tropical climate, represents a favorable growth zone for the olive tree [2]. To date, the main producer of EVOO is Spain (37%), followed then by Italy (24%) [3]. EVOO is produced from the fruit of the olive tree (*Olea europaea*), through mechanical or physical processes including washing, decanting, centrifugation and filtration. An EVOO to be defined as such must be produced through these processes and have a number of organoleptic characteristics [4]. EVOO has nutritional and organoleptic characteristics, due to the presence of two fractions, the one containing the major components (triglycerides, diglycerides and free fatty acids) and the one containing the minor components (pigments, tocopherols and phenolic compounds) [5]. EVOO has a high content of monounsaturated fatty acids, mainly oleic acid, and some polyunsaturated fatty acids such as linoleic acid. Regarding minor components, their main representatives are phenolic compounds, for instancehydroxytyrosol and derivatives (oleuropein and tyrosol), tocopherols but also other compounds such as hydrocarbons (e.g., squalene) or pigments such as provitamin A compounds. It is interesting to underline that the concentrations of minor and major components in oil depend on the ripening process of olives, atmospheric and agricultural conditions, genetic control and extraction process [6]. For example, during the ripening process, as the content of oil inside the drupes increases, the photosynthetic activity decreases. In fact, in the first phase of ripening (green phase), the ripe fruits have already reached their final size, therefore ripening continues, and the chlorophylls present in the skin are gradually exchanged with anthocyanins, transforming the fruits from green to dark purple or violet until harvesting. These color changes result in spotted, purple and black olives. Generally, olives have the highest content of phenolic compounds in the phase from green to darker skin. For the quality of EVOO, it is important to choose the correct harvesting time, identifying the optimal ripening stage that may vary according to region of production and cultivars milled [7].

Other compounds that play an important role in keeping under control the genuineness of EVOO are sterols specifically four important classes: 4-Desmethylsterols, 4α-Methylsterols, triterpene alcohols (4,4-Dimethylsterols), and triterpene dialcohols. Two isoforms of tocopherols are also present in EVOO: α-and γ-tocopherol. Generally, the presence of tocopherols is influenced by both the years of harvest and the distance between plants. In addition, if high concentrations of tocopherols are present, there are also high concentrations of chlorophyll components [8].

The main cluster of antioxidants present in EVOO are the hydrophilic phenols; they allow defining the quality of the oil on the bases of their sensory characteristics, such as bitterness, pungency and stability [9]. Secoiridoid compounds are responsible for the bitterness of the oil. Finally, the carotenoids and chlorophyll components present in EVOO affect its characteristic color. The color of EVOO is greener in the presence of green olives that have a higher content of chlorophyll components, while using ripe olives with a superior carotenoid content results in a more yellowish oil, so the proportions of these pigments influence the final color [7].

Generally, the sensorial characteristics of EVOO mainly depend on the state of maturity of olives, according to their color. In fact, green olives contain a large number of ingredients with a strong flavor of fruit. Indeed, from ripe olives it is possible to obtain a higher yield than from early maturing stages. Moreover, when olives ripen, the number of aromatic elements decreases. This is the reason why EVOO derived from purple and black olives has a softer taste and a milder smell [10]. From green olives, a stable oil is obtained (due to the antioxidant effect of phenols) with a high phenol content which affects both bitterness and pungency. The highest concentration of aromatic compounds occurs when the fruit begins to change color, and then decreases as the olive matures. In fact, in oils made from ripe olives the so-called “fat fruitiness” is at lower concentrations than in oils made from green olives. As a result, the sensory notes of bitter-pungent and other aromatic sensations decrease as the degree of ripeness increases [11]. Despite this, sometimes early harvesting allows the production of oils that are often not organoleptically acceptable due to excessive phenols concentrations [12]. Although the chemical composition of the oil is a valid method to identify the characteristics and quality of an oil, also panel test represents a crucial tool in its characterization. This analysis allows to standardize the evaluation procedures of the organoleptic attributes of oils and to establish specific quality classes, namely extra virgin olive oil (EVOO), virgin olive oil (VO), ordinary virgin olive oil (OO) and lampante olive oil (LO). This approach makes use of a group of evaluators (or panelists) properly trained to distinguish and rate the presence of positive and negative characteristics. In this regard, the International Olive Council (IOC) has provided standards for the sensory test of oils. In this way, tasters have guidance regarding the proper conditions of the sensory laboratory, the tasting conditions and characteristics of the glass for the organoleptic analysis of oils, as well as the sensory procedure and rules for the classification, training and monitoring of expert tasters of virgin olive oil [13]. In panel testing, a group of panelists (8 to 12 experts) subjectively evaluate EVOO samples (with numbers from 0 to 5) through tasting. Naturally, this approach uses subjectivity as a measuring tool. Subsequently, it is eliminated by statistical methods applied to all panelists’ scores and the values obtained, for each question [14]. 

As reported above, the quality of EVOO is strongly influenced by several factors, such as the condition of the raw material. However, this last can undergo a physical selection process to differentiate the final production quality. Salvucci et al. [15] proved how RGB image processing systems can offer a support to the production of high-quality goods, classificating automatically and quickly olive lots into different qualitative classes, for both oil and table olive production. Nowadays, this selection process can be achieved using machines capable of selecting large batch of fruits on the base of real-time image analysis. This system allows an objective and fast selection otherwise impossible by hand. Even if optical sorting machines are widely used for the selection of different types of fruit and vegetables for different scopes, olive sorting is an emerging process (in the EVOO production chain) to improve the quality of the final product. Sorting machines are generally equipped with high resolution cameras that may operate in different spectral ranges, electronic control unit o (ECU) and processing unit, conveyor belts or chutes, and devices to physically separate the product on the base of the analysis results (e.g., pneumatic systems).

Many studies found in literature regard the evaluation of image analysis algorithm to select the olives without a real physical selection test. For example, Puerto et al. [16] have classified olives, harvested on the plant or from the ground, following the roughness of the surface and the color of the fruits. Also, Babanatis-Merce et al. [17] developed an image analysis classifier to distinguish olives on the base of their color. Very few works tested proper physical systems, even at small lab scale, to select olives. Some investigated the selection in lab of small amounts of olives at low speed separating one olive per time [10,11,12,13,14,15,16,17,18].

The literature does not show any work regarding olives industrial selection for EVOO production. In addition, no work was found regarding separated milling of each classified olives lot and their physico-chemical and sensorial characterization.

Therefore, the aim of this work was to evaluate for the first time the performances of an industrial RGB optical sorting prototype for the classification of olives into two classes, on the base of their maturity and the presence of external defects. The performance of such classification has been evaluated comparing the results with those obtained visually by a trained operator. Moreover, the olives divided into three classes (named “Mixed”, “First choice” and “Second choice”) have been milled and the obtained EVOOs were analyzed by physico-chemical, chemical and sensory analyses.

## 2. Materials and Methods

In this work, two different olive cultivars were considered (i.e., Salviana and Leccino). Olives freshly harvested were provided by Narducci Oil Mill, located in Moricone (Central Italy) and member of the SABINA DOP Consortium. The two cultivars are mainly present in the territory of Sabina. On 2nd November (2021 season) 929 kg (gross) of olives were harvested, including twigs and wet leaves.

After a manual gross cleaning, the net weight was 845 kg. On 4th November tests were carried out, using an RGB optical sorter, and milling followed.

### 2.1. Optoelectronic Sorting Machine Olive Classification

The selection of olives was carried out using an optoelectronic pneumatic sorting machine INFINITY Plus (Italian Sorting Technology, I.S.T. Srl, Ferrara, Italy) (Figure 1) adapted for the scope.

This sorting machine allows to identify both the characteristics and the defects of the olives through digital trichromatic (RGB) cameras, located in anti-parallel position, that distinguish up to 16 million colors and defects down to 0.09 mm.

The INFINITY Plus model allows the selection of small products and is equipped with one chute to slide and guide the items to be selected. The machine uses a software that allows the user to manage it without the help of a technician by selecting the products to be retained and discarded through a preventive programming. In details, the machine uses a supervised approach, indeed prior selection a manual training is needed. At the beginning the operator must choose samples owing to the “Bad”, “Good” and “Spot” classes that need to be used as initial image acquisition for proper training and calibration. The “Spot” class determines a specific defect, in particular the presence of large spots on green olives. Once images are acquired, the operator visualize these on the machine screen and set up the best options creating the “recipe” that better identify the wanted classes specifying a precise threshold for each of the single R, G and B channels values. Moreover such “recipe” includes for each of the named channel the n. of pixel per each threshold range that should be considered for the wanted classes. Following this principle, after initial training, the recipe can be saved, used, and recalled as needed later on. The whole image processing flow consists of the initial image acquisition, during the item fall, the real-time image segmentation for background subtraction, and segmentation of each single item on the base of the indications written in the recipe loaded for processing.

The machine characteristics are presented in Table 1 below. Please note that the max selection amount is referred to cereals.

The sorting machine was used to physically classify the initial batch (Mixed choice) into two separated classes (named First choice and Second choice)

As for the general settings, the “Reject” has been set, i.e., the sensitivity of the selection, to a value of 2000 pixels, for the “Bad” class. For the “Spot” field a threshold of 2600 pixels (9 mm^2^) has been set. Only RGB cameras, positioned both in front and behind, have been activated. In this screen you can also set the speed of the vibrators in a range between 0 and 100, to run the product on the slides. The images captured by the class can be seen in the debug screen to see that the cameras are working properly. The parameters of the solenoid valves that control product ejection then were managed. The following parameters were entered in the solenoid valve settings screen: the time the solenoid valve remains open, which is a value of 40 in units of 100 micro seconds; the delay, which is the time between the detection of the defect and the opening of the solenoid valve, set to 10 in units of 100 micro seconds; the burst, the number of pixels that are added to the defect to determine how many solenoid valves need to be opened at once, set to 3 pixels. The detected images can be stored using the “Capture” button. Each image can be modified according to the values “Green”, “Red” and “Blue”, differentiating for each class the values to increase, to isolate the color of the defect, or decrease, to discount. When the parameters have been set, the classes are created, and the process is repeated on each camera. The sorting machine was set at two different speeds: slow (vibrators was set to 30%) and fast (55%).

### 2.2. Visual Evaluation

Two classes of olives were identified on the basis of ripeness (“green”, those with an early maturation stage, and “black”, those ripen). In addition, we also considered the olives invaiate in the class “mixed”. The batches of olives were sampled manually, dividing them into the 2 types in order to measure the efficiency of the sorting machine (Figure 2).

### 2.3. EVOOs Production and Chemical-Physical, Chemical Analyses

On the other hand, as far as milling is concerned, the mill uses a stainless-steel Alfa Laval (Monza, Italy) system with toothed disks at low speed to avoid overheating and improve the sensory characteristics and quality of the dough. Kneading takes place for 40 min in stainless steel tanks at controlled temperature (26/27 °C). In addition, the system uses the two-phase decanter Alfa Laval X7.

For each oil, in duplicate, were performed some quality analyses according to the European Union Commission Regulation EEC/2568/91 and its subsequent amendments: free fatty acid, peroxide index, UV indices (K232, K270, ΔK), fatty acid ethyl esters (FAEEs) [19]. In addition, following the Cucurachi protocol, were measured the VIS absorptions of the pigments: chlorophyll and carotenoid [20]. The determination of the tocopherols was performed by HPLC method [21]. About content of the phenolic compounds, it was applied the protocol explained in the COI/T.20/Doc No 29/2009 [22].

### 2.4. Organoleptic Characteristics

The organoleptic profile of the olive oils was obtained under the conditions stated in the European Union Commission Regulation EEC/2568/91 and its subsequent amendments (Annex XII) by official panel recognized by the International Olive Oil Council (IOC) and the Ministry of Agricultural, Food and Forestry Policies (MiPAAF). Each panel taster smelled and tasted the oil considered, following the profile sheet provided by the Annex XII and to COI/T.20/DOC. 22-2009 [22]. The panel test inspected the presence of the attributes: fruity, bitter, pungent, grass, almond, artichoke, aromatic herbs and tomato.

### 2.5. Statistical Analyses

The attributes were evaluated in a defined range from 0.0 to 10.0 and analyzed to assess the median while confidence intervals were calculated considering, as significative, a coefficient of variation equal or lower than 20%.

A statistical ordination approach has been applied to the matrix of the oil duplicates of more significative analytical variables and the oils coming from different rates of green olives. In particular a principal components analysis (PCA) has been conducted using the software PAST [23].

## 3. Results

### 3.1. Olive Classification Based on Sorting Machine

The quantities of oil produced by the selected olives are shown below in Table 2. As shown in Table 2, a higher oil yield was obtained from ripe olives (20.40 kg).

The yield of an oil is obtained from the physical extraction of olives (25% of the initial weight, generally). To obtain a higher yield, the olive harvest must be done when the drupes are ripe and therefore able to give the best oil yield [24]. In general, for the purposes of oil quality, the optimal period of harvesting is deeply influenced by the degree of ripeness of the drupes. The harvest should be carried out when the olives are completely invaiate and give a high yield in oil in relation to the fresh weight [25]. The classification carried out with the sorter has yielded the following percentages of error for the two classes (“green” and “black”), as reported in the Table 3.

### 3.2. Chemical-Physical, Chemical, Sensory Profile and Statistical Analyses

Based on the quality indices stated in the EC Regulation [19], the three kinds of oil, giving from olives at different stage of ripeness, were idded as extra virgin (EVOOs); below the oils coming from unselected olives with 61.5% of green olives will be called EVOOMix (Mixed), while the oils coming from optical selection of olives will be identified with the acronyms EVOOFC (First Choice) with 88.2% of green olives, and EVOOSC (Second Choice) with 34.7% of green olives. A complete characterization of the investigated EVOOs is reported in Table 4 and Table 5.

The samples analyzed showed values of free acidity, as percentage of oleic acid, in a narrow range between 0.44 for EVOOFC and 0.28 for EVOOMix; while, about peroxide values they were comprised in the interval: 5.3 for EVOOFC and 4.1 for EVOOSC. EVOOFC showed the minimum values Κ232, Κ270, and ΔΚ that were respectively 1.673, 0.097 and 0.003. EVOOMix and EVOOSC had higher values, in any case compliant with legal values. As for the substances responsible of color, EVOOFC showed the higher values of absorptions of carotenoids (400–500 nm), and chlorophyllides (500–700 nm); middle absorption values are shown by EVOOMix; lower values, instead, are shown by EVOOSC.

All samples showed FAEE content well below the official value of 35 mg kg^−1^ for extra virgin, even if EVOOSC presented of 8.5 mg kg^−1^, proving the good life and oil quality, obtained by appropriate agronomic and technological practices; avoiding damaging fermentation and degradation phenomena [26,27]. About waxes, the values were clouded from 28.4 for EVOOMix to 38.8 mg kg^−1^ for EVOOSC, as for oil of category “extra virgin”. Another indicator of good olive processing practice and storage conditions is the content of the tocopherols, within the range 234.5 for EVOOSC and 286,4 mg kg^−1^ for EVOOFC, desirable also for his high nutritional value. The phenols are substances with a strong action [28,29], giving a distinctive taste [30,31] related to different factors as cultivar and technologies, as expected for the oils obtained by more ripe olives, EVOOSC showed the lower content of total phenols (454.0 mg kg^−1^), of secoiridoids (365.8 mg kg^−1^) and of total phenols without oxidized forms (412.3 mg kg^−1^), while the total phenols content without oxidized forms was medium: 413.3 mg kg^−1^ for EVOOSC, 433.6 mg kg^−1^ for EVOOMix and 438.5 mg kg^−1^ for EVOOFC.

Through organoleptic analysis mentioned in the official method [19], all oils were listed as extra virgin, with the median of defects equal to 0.0 and the median of fruity greater than 0.0. The organoleptic profile of the EVOOs investigated in shown in Figure 3.

The profiles were characterized by harmonized intensity of attributes of fruity, bitter, and pungent, although EVOOSC showed the lower content of all evaluated sensory attributes, in agreement with a large study, where was demonstrated that the mentioned attributes are quantitatively reduced in all kinds of olives, during the grow ripe of the fruits, with differences between different cultivars [32]. EVOOFC showed on average the highest intensities of pungent (5.15), fresh almond (4.45) and artichoke (4.45) sensation.

## 4. Discussion

### 4.1. Olive Classification Based on Sorting Machine

The study shows satisfactory results with error percentages in olive classification, equal to 13% for green ones and 36.3% for the black ones (Table 3). In the latter case the higher error rate depends on the classification of spotted and bruised olives as black. In Babanatis-Merce et al. [17], the percentage of correct classification in the selection of green and black olives was between 67% and 86%. In this case, however, a color recognition software was used on a smaller quantity of olives without using an industrial optical sorter whereas in the work of Puerto et al. [16], the classification of olives through an artificial intelligence algorithm, showed 100% correct classification. Therefore, this study presented in this article represents a technological innovation in the automatic classification of olives, also considering the correct and acceptable classification reported, on par with, and perhaps even superior to, other technologies commonly used in this sector nowadays. Indeed, today, there are no works that analyze and optimize industrial optical sorter operation for olive classification.

### 4.2. Chemical-Physical, Chemical, Sensory Profile and Statistical Analyses

After the application of pattern recognition technique using unsupervised Principal Components Analysis (PCA) on the all-analytical variables from chemical-physical, chemical analyses and the panel test attributes, descriptors with loadings on the first three principal components close to zero were excluded from the statistical treatment. Then the biplot of the final principal component analysis was obtained. It considered the most significative analytical variables—A414, A446, A474, A532, A664, R A474/A664, Tocopherols, Ethyl esters, Waxes, Secoiridoids, Total phenols-Oxidized forms, Fruity, Bitter, Pungent, Grass, Almond, Artichoke—as reported in Figure 4.

It showed along the first component the cluster of variables (a): FAEE, Waxes and R A474/A674 more connected to samples of EVOOSC oil obtained by olives in an advanced state of ripeness, with 34.7% of green olives; along the second component other two clusters of the variables (b): A414, A446, A474, A532, A664, R A474/A664, Pungent, Artichoke and Almond more connected to EVOOFC obtained by less ripe olives, with 88.2% of green olives and, (c): Tocopherols, Phenols, Secoiridoids, Fruity, Bitter and Grass more connected to EVOOMix, obtained by middle ripe olives, with 61.5% of green olives. Therefore, the biplot of the PCA explains the relation between clusters of the most relevant analytical variables and oils composition, depending on the green olive rate.

## 5. Conclusions

Digital technologies in general and optoelectronic ones in details represent a mature and solid reality implemented in many food chains to physically classify products on the base of their qualitative characteristics. The advantages coming from their implementation regard the objectivity of the assessments, the processing speed, the capacity to work (potentially) without interruption, unlike an operator, etc. In the EVOO sector these devices are being implemented lately to select the olives in order to produce oil with “superior” qualities or e.g., to eliminate defected olives. The results produced by the present work show for the first time the potential of these technologies testing an industrial system on olives for EVOO production. The analyses the oils obtained show significative differences both chemically and through the panel tests carried out. This characterization represents important potential benefits. The selection gives the possibility to carry out late harvest still being able to obtain oils with characteristics desired as those represented by oils milled from unripen fruits. Indeed, the present work was carried out at the mid-end of the olive oil campaign when both, not ripen and completely ripen fruits are contemporary present. Moreover, it gives the opportunities to eventually separate, with proper calibration and training, defected olives. In addition, the Italian olive farming systems very often present small size and several olive tree varieties. These are mixed in the field and thus are commonly contemporary harvested even if presenting different ripening stages. A selection potentially rises the opportunity to even blend the oils obtained to create different products following the consumer or the producer’s needs. Obviously, technologies come at a cost that could be faced by a medium size mill or at consortium level to which the small olive producer confers. Future work will test and evaluated the included Nir sensor potential. Moreover, since a degradation based only on the color classification of input olives could depend also on the percentage of certain defects that could not be seen through RGB analysis (e.g., those on black olives), in the future work, these will be precisely determined prior selection for later on consideration.

## Figures and Tables

**Figure 1 foods-11-02917-f001:**
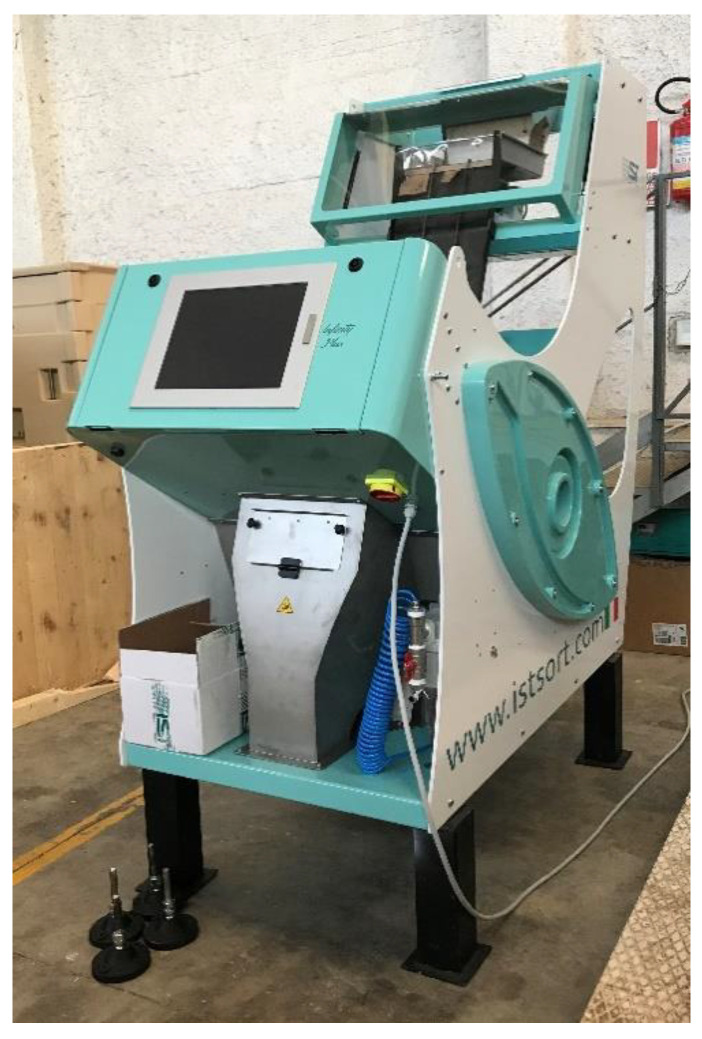
INFINITY PLUS sorting machine.

**Figure 2 foods-11-02917-f002:**
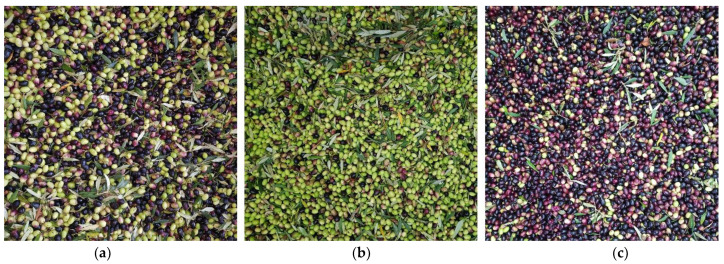
(**a**) “mixed”, (**b**) “green” and (**c**) “black” olives classes.

**Figure 3 foods-11-02917-f003:**
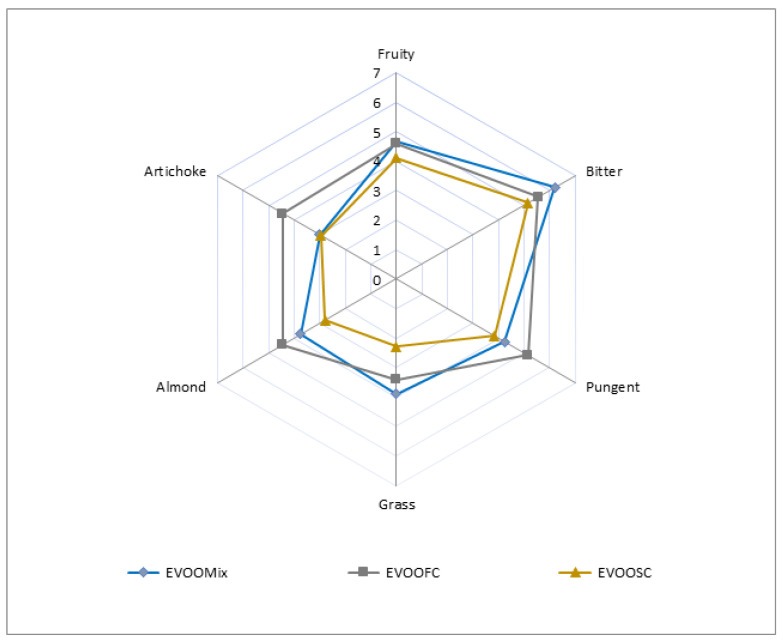
Sensory profile of the Mixed, First Choice and Second Choice EVOOs analyzed, i.e., EVOOMix, EVOOFC and EVOOSC respectively.

**Figure 4 foods-11-02917-f004:**
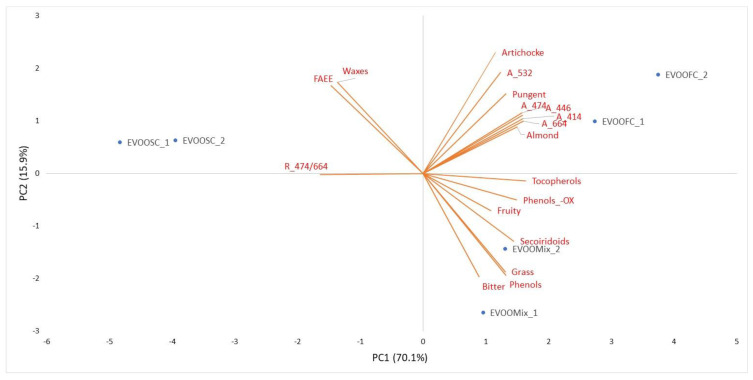
Biplot of the PCA conducted on of some chemical-physical, chemical analyses and the sensory profile attributes.

**Table 1 foods-11-02917-t001:** Sorting machine characteristics.

Characteristics	INFINITY 1 Plus
W × D × H	846 ∗ 1774 ∗ 1825
Air consumption (L/s)	8
Power consumption (kW)	0.494
Max RGB resolutions (mm)	0.09
N. RGB camera	2
Camera color sensitivity (million)	16
N. Nir camera	1
Max selection amount (t/h)	4

**Table 2 foods-11-02917-t002:** Weight of processed olives and produced oil and relative yield. The first class EVOO is obtained from GOOD selected olives, the second class EVOO from BAD selected olives and mixed EVOO from unprocessed mixed olives.

EVOO	Weight of Selected Olives (kg)	Weight of Produced EVOO (kg)	Yield (%)
First class	256	31	12.10
Second class	299	61	20.40
Mixed	272	42	15.44
TOTAL	845	135	47.94

**Table 3 foods-11-02917-t003:** Error rate in olive classification.

Classification	Correct Classification (%)	Error (%)
Green	87.0	13.0
Black	63.7	36.3

**Table 4 foods-11-02917-t004:** Chemical-physical and chemical results of the analysed EVOOs.

Parameters	EVOOMix_1	EVOOMix_2	EVOOFC_1	EVOOFC_2	EVOOSC_1	EVOOSC_2
Free acidity (% oleic acid)	0.29	**0.28**	**0.44**	0.42	0.36	0.33
Peroxide value (meqO_2_ kg^−1^)	4.3	4.4	**5.3**	**5.3**	**4.1**	4.4
K_232_ (max 2.50)	1.691	**1.709**	1.673	**1.688**	1.708	1.689
K_270_ (max 0.22)	0.101	**0.105**	0.097	**0.099**	**0.105**	**0.105**
∆K (max 0.01)	−0.004	−0.004	**−0.003**	−0.004	−0.004	−0.004
A_414_	0.851	0.899	1.060	1.050	0.717	0.740
A_446_	0.794	0.828	0.987	0.982	0.680	0.698
A_474_	0.652	0.685	0.808	0.806	0.558	571
A_532_	0.041	0.040	0.049	0.061	0.041	0.035
A_664_	0.279	0.298	0.357	0.353	0.226	0.236
R A_474_/A_664_	2.341	2.300	2.263	2.284	2.465	2.414
Tocopherols (mg kg^−1^)	270.6	269.6	**286.4**	279.5	239.7	**234.5**
Ethyl esters (mg kg^−1^)	3.5	**3.0**	4.6	4.3	**8.5**	7.6
Waxes (mg kg^−1^)	30.0	**28.4**	32.0	31.5	36.6	**38.8**
Total phenols (mg kg^−1^)	**490.4**	487.7	488.1	471.9	**454.0**	457.8
Secoiridoids (mg kg^−1^)	**400.1**	387.5	398.9	388.5	369.2	**365.8**
Low molecular weight (mg kg^−1^)	**3.4**	4.3	4.6	5.1	**7.0**	6.0
Medium molecular weight (mg kg^−1^)	197.8	195.5	**232.4**	227.2	181.9	**178.5**
Ox-forms (mg kg^−1^)	**42.5**	37.1	**24.2**	26.1	30.4	31.5
High molecular weight (mg kg^−1^)	**198.9**	187.7	161.9	**156.2**	180.4	181.3
Oleuropein (mg kg^−1^)	**101.5**	92.8	**63.3**	68.7	86.8	81.9
Ox-forms (mg kg^−1^)	16.0	15.2	**17.7**	15.0	**11.3**	12.3
Lignans (mg kg^−1^)	**58.0**	63.7	57.8	**54.4**	55.1	**58.0**
Other phenols (mg kg^−1^)	31.6	**32.6**	**27.1**	25.5	27.8	30.1
Total phenols-Ox-forms (mg kg^−1^)	431.9	435.3	**446.3**	430.8	412.3	**414.0**

**Table 5 foods-11-02917-t005:** Sensory profile of the analysed EVOOs.

Sensory Evaluation	EVOOMix_1	EVOOMix_2	EVOOFC_1	EVOOFC_2	EVOOSC_1	EVOOSC_2
Fruity	4.9	4.4	4.2	5.0	3.8	4.4
Bitter	6.3	6.1	5.0	6.1	5.1	5.2
Pungent	4.3	4.2	4.5	5.8	3.8	3.9
Grass	4.3	3.5	3.1	3.7	2.1	2.5
Almond	3.2	4.3	4.1	4.8	2.5	3.1
Artichoke	2.7	3.3	3.9	5.0	2.5	3.4

## Data Availability

All data are included in the main text.

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
