# Peer review of "Superior EVOO Quality Production: An RGB Sorting Machine for Olive Classification"

_foods, 2022, doi:10.3390/foods11182917_

Round 1

Reviewer 1 Report

* Lines 27 and 131: The claim "“For the first time, the optoelectronic technologies in 

an industrial system was tested on olives to produce superior quality 28 EVOO”  is not accurate. 

See for instance European project :

https://cordis.europa.eu/project/id/811930

* Please review spelling. For instance, in line 198 "invaiate". In line 164 "owing->belonging"?,...

* Table 2: totals are inconsistent. Please review the numbers.

* General comment: the olives categories  are not clearly named through the paper. 

  - At the beginning of Materials and Methods section, they mention ripeness and defects criteria to sort into three categories "bad", "good" and "spot".  

  - Then, authors talk about classes ("Mix choice", "first choice", "second choice". 

  - Then in Figure 2 we can see "mixed", "green" and black and in table2 "first class", "second class", and "mixed". 

  Some sort of unification would be desired.

* Heading of table 4. Review line breaks, for instance EVOOMix_1

* Little detail is given about the computer vision sorting methods.

* General Opinion. The paper is somewhat dense, specially the materials and methods section. 

Too many numbers, and sure all of them are relevant for the conclusions of the paper. 

The results obtained are not particularly relevant neither in classification efficiency nor in proving the relation of classificacion with quality. The only clear conclusion is that there are  slight 

differences in organoleptic features, mainly due to:

  - Ripness classification is done using color but there is a range of ripeness that can not be 

  observed by color.

  - Ripeness index of olives is not related only with color. When the olive skin is black, there may be different degrees of maturity  and an overripe olive is as dark as an olive hat has just turned from green to black. The only way to tell the differences (as far as I know) is to use other wavelengths such as infrared.

  - It is very difficult that chemical-physical parameters show a degradation based only on the color classification of input olives. The oil quality depends more on the presence with a high enough percentage of certain defects.

  - Another limitation is that defects can only be detected on green olives. Dark defects can not be detected in black olives.

  - The efficiency of the classification is not very good as can be seen in the % of errors. This makes that there is a significant mix at the different outputs, and the results can not be se clear.

Author Response

Comments and Suggestions for Authors

1) Lines 27 and 131: The claim “For the first time, the optoelectronic technologies in an industrial system was tested on olives to produce superior quality 28 EVOO” is not accurate. See for instance European project :https://cordis.europa.eu/project/id/811930

In the European Project you pointed out to us, there is no mention of an industrial sorter, but rather of a sorter that could represent a breakthrough in technology but is still patent pending. Instead, the optical sorter used in our work is already for sale and in use in industrial companies.

2) Please review spelling. For instance, in line 198 “invaiate”. In line 164 “owing->belonging”?,...

We reviewed words in the text. We replaced “invaiate” with “ripened” and “owing” with “relating” as suggested to reviewer 3.

3) Table 2: totals are inconsistent. Please review the numbers.

We reviewed and modified the numbers.

4) General comment: the olives categories are not clearly named through the paper.

  • At the beginning of Materials and Methods section, they mention ripeness and defects criteria to sort into three categories “bad”, “good” and “spot”.
  • Then, authors talk about classes “Mix choice”, “first choice”, “second choice”.
  • Then in Figure 2 we can see “mixed”, “green” and black and in table2 “first class”, “second class”, and “mixed”.
  • Some sort of unification would be desired.

At the beginning of the M&M section, three criteria are mentioned, namely “Good”, “Bad” and “Spot” which refer to the olive grading method used by the grader during operation. The classes “Mix choice,” “First choice” and “Second choice” correspond to those shown in Figure 2 and Table 2. As you suggested, we have unified the classes all under one name (First choice, Second choice and Mixed choice).

5) Heading of table 4. Review line breaks, for instance EVOOMix_1

Thanks for your comment. We reviewed line breaks.

6) Little detail is given about the computer vision sorting methods. FEDERICO

Additional details are now present in the text.

7) General Opinion. The paper is somewhat dense, specially the materials and methods section. Too many numbers, and sure all of them are relevant for the conclusions of the paper. The results obtained are not particularly relevant neither in classification efficiency nor in proving the relation of classificacion with quality. The only clear conclusion is that there are slight differences in organoleptic features, mainly due to:

  • Ripness classification is done using color but there is a range of ripeness that cannot be observed by color.
  • Ripeness index of olives is not related only with color. When the olive skin is black, there may be different degrees of maturity and an overripe olive is as dark as an olive hat has just turned from green to black. The only way to tell the differences (as far as I know) is to use other wavelengths such as infrared.

Thanks for the comment we are aware of that and taking this in mind we wanted to verify what was the output of this kind of instruments that are being sold. However, since the machine include an IR camera, we plan to deepen the study including the IR acquisition and processing during the project (this was already present in the conclusion).

  • It is very difficult that chemical-physical parameters show a degradation based only on the color classification of input olives. The oil quality depends more on the presence with a high enough percentage of certain defects.
  • Another limitation is that defects can only be detected on green olives. Dark defects cannot be detected in black olives.

Again, thank you, this consideration has been added in the conclusion of the manuscript now.

  • The efficiency of the classification is not very good as can be seen in the % of errors. This makes that there is a significant mix at the different outputs, and the results cannot be se clear.

Thank you, we do hope to significantly improve the output through the combined use of RGB and infrared sensors, however there are already discrete differences in the final oils evaluated.

Reviewer 2 Report

The olives ripeness and the choice of harvest time according to their color and size, strongly influences the quality of the EVOO. The physical sorting of olives with machines performing rapid and objective optical selection, impossible by hand, can improve the quality of the final product. The authors described that the classification of olives into two qualitative classes, based on the maturity stage and the presence of external defects, through an industrial RGB optical sorting prototype, evaluating its performance and comparing the results with those obtained visually by trained operators. That work is interesting and very useful for olive oil quality-improvement. However, some revisions should be made before publication. They are showed as follows.

Comments:

1) In the part of “Discussion”, the discussion the authors gave is not deep, and is mainly lack of comparison with other reported literatures.

2) Previous literature “Olive classification according to RGB, HSV and L*a*b* color parameters using Image processing technique” have showed that the defected and sound olives by image processing technique could be discriminated. What is new or innovation of your workplease be stated.

3) The description on the cutting-edge technique concerning RGB sorting of this work in Introduction seems not enough.

4) Visual evaluation in Line 196: the sorting rules of olive ripeness (green, mix and black) are not explicit. A quantification rules should be suggested.

Author Response

Comments and Suggestions for Authors

The olives ripeness and the choice of harvest time according to their color and size, strongly influences the quality of the EVOO. The physical sorting of olives with machines performing rapid and objective optical selection, impossible by hand, can improve the quality of the final product. The authors described that the classification of olives into two qualitative classes, based on the maturity stage and the presence of external defects, through an industrial RGB optical sorting prototype, evaluating its performance and comparing the results with those obtained visually by trained operators. That work is interesting and very useful for olive oil quality-improvement. However, some revisions should be made before publication. They are showed as follows.

Comments:

1) In the part of “Discussion”, the discussion the authors gave is not deep, and is mainly lack of comparison with other reported literatures.

The discussion has not been deepened by referring to other publications because there is not much literature on the subject (as cited in the introduction); in particular, the sorters that are used are not industrial as in our work and consequently comparisons cannot be made.

2) Previous literature “Olive classification according to RGB, HSV and L*a*b* color parameters using Image processing technique” have showed that the defected and sound olives by image processing technique could be discriminated. What is new or innovation of your work? please be stated.

The innovations related to this work are not only about sorting olives by discriminating between defective and healthy olives using image processing technique. The work aims to produce a higher quality EVOO by eliminating defective olives from selection. In fact, the panel test performed on “First choice,” “Second choice” and “Mixed choice” olives also reported different results (this topic has already been expressed in the discussion and conclusion).

3) The description on the cutting-edge technique concerning RGB sorting of this work in Introduction seems not enough.

As suggested two references regarding the use of RGB associated with EVOO have been added in the Introduction. Regarding RGB sorting, no more was found other than the one already mentioned in the article, as there are not many works on this subject. Consequently, this work of ours represents a novel approach using an RGB chamber applied to an industrial, non-laboratory sorting machine.

4) Visual evaluation in Line 196: the sorting rules of olive ripeness (green, mix and black) are not explicit. A quantification rules should be suggested.

The quantification rule is expressed in lines 289-294: “The yield of an oil is obtained from the physical extraction of the olives (25 percent of the initial weight, generally). To obtain a higher yield, olives should be harvested when the drupes are ripe and thus capable of giving the best oil yield [24]. In general, for the purposes of oil quality, the optimal harvesting period is strongly influenced by the degree of ripeness of the drupes. Harvesting should be done when the olives are fully versioned and give a high oil yield relative to fresh weight [25].” Given this statement related to the quality of EVOO, three classes were identified based on olive ripeness.

Reviewer 3 Report

Line 36   About 82 % 

Is reference 3 correct as I could not find reference to % of production in Spain and Italy. If not please correct.

Line 41 you refer to mechanical methods yet  do not list a mechanical method

Line 42  For an EVOO to be defined as such it

Line 45  and the other containing the

Reference 6  - are you sure this is correct. Please go through the first part of the reference section and make sure that the references do indeed say what you quote.

Line 60 in the phase between the green and

Line 62 may vary according to the production region and cultivars milled

Line 63 the genuiness of the

Line 71 allow the determination of the quality of the oil with respect to sensory characteristics

Please check with editor if Food uses US or British spelling. If British then correct colour, flavour

Line 80 ingredients with a strong fruit flavour.

Line 93 oil, panel testing represents  a crucial

Line 93 This allows the standardisation of the evaluation procedures

Line 156  allows the identification of both

Line 162 In detail

Line 163 indeed prior selection manual training is needed.

Line 164 beginning the operator must choose samples relating to the

Line 168 and sets up

Better identifies

Line 170 the programme can be saved

Line 169 Are you sure n. of pixels makes sense? What is n?

Line 172-173   max in full?

Line 173 Selection amount refers to cereals

Line 218 The VIS absorptions yielding

Line 219 the Cucurachi

Line 224 was set at 80

Line 228 a solution (remove was prepared before a solution)

Line 229                        1 g of

Line 237 Place The phenolic compounds in a new paragraph

Line 177 With respect to the general settings, the Reject choice was set as follows, ie

Line 179 in front and behind, were activated

Line 180 On this screen it was possible to set the speed

Line 184 the solenoid valve remained

Line 188 valves need to the opened at once was set

Line 190 re-write lines 190 to 194

Line 195 -198 re-write could not follow

Figure 2 and text: Why does it refer to 2 choices when there are in fact 3?

Clarify

Line 204 Delete on the other hand as far as milling was concerned.

Instead pls use : The mill used a stainless-steel Alfa

Line 206 Kneading took place for 40

Line 208 used a two-phase

Line 271 Are you sure the word is leave – do you refer to leaf or leaves?

Line 276 A statistical ordination approach was

Line 276 not clear

Line 278 olives. A principal component analysis (PCA) was conducted using the software

Table 2 preferably do not split the word olives

Line 301  obtained from the olives at different

Line 302 green olives were called EVOOmix

Line 303 olives were identified

Line 311  if you refer to a range should you not show highest and lowest figures?

Line 312 same argument as above

 Line 317 400 and 500 nm, that correlated to the concentrations of carotenoids and between 500 and 700 nm that corresponded to

Line 323 Did not follow , possibly re-write

Line 333 for the oils obtained from more ripe olives, EVOOSC showed a lower

Line 348 the lowest concentration

Line 349 where it was demonstrated

Line 355 The study showed

Line 357 error rate depended

Line 362 Therefore, the study

Line 370 After the application of the pattern recognition technique, an unsupervised Principal Component Technique (PCA) was applied to the……

Line 372 A biplot of the final PCA was then obtained.

Figure 4 legend Biplot of the PCA conducted on some of the chemical

Line 380 Along the first components

Line 382 – 385 please clarify sentence

Line 390 and optoelectronic ones in particular represent important tools implemented

Line 393 of the assessments, processing speed, capacity to work, (potentially) without interruption etc.

Line 396 to eliminate defective olives

Line 397 the potential of these technologies for use in EVOO production.

Line 398 The oil analyses obtained showed significant differences both chemically

Line 401  while still being able

Line 403 when both unripened and completely ripe

Line 407 thus are commonly concurrently harvested

Line 408 A selection potentially gives the opportunity

Line 411 Future work will test and evaluate 

Author Response

Reviewer #3

Line 36 About 82 %

As suggested, we modified the text.

Is reference 3 correct as I could not find reference to % of production in Spain and Italy. If not please correct.

The reference 3 is correct.

Line 41 you refer to mechanical methods yet do not list a mechanical method

Reference is made to mechanical methods because the processes for obtaining EVOO (mentioned in line 41) are mechanical processes (washing, decanting, centrifugation, and filtration).

Line 42 For an EVOO to be defined as such it

As suggested, we modified the text.

Line 45 and the other containing the

As suggested, we modified the text.

Reference 6 - are you sure this is correct. Please go through the first part of the reference section and make sure that the references do indeed say what you quote.

As suggested, we rechecked the references. They are all corrects.

Line 60 in the phase between the green and

As suggested, we modified the text.

Line 62 may vary according to the production region and cultivars milled

As suggested, we modified the text.

Line 63 the genuiness of the

As suggested, we modified the text.

Line 71 allow the determination of the quality of the oil with respect to sensory characteristics

As suggested, we modified the text.

Please check with editor if Food uses US or British spelling. If British then correct colour, flavour

Thank you for your comment. The journal guidelines state that either English or UK English can be used as long as there is consistency. Indeed, we replaced “color” with “colour” and “flavor” with “flavours” in all text.

Line 80 ingredients with a strong fruit flavour.

As suggested, we modified the text.

Line 93 oil, panel testing represents a crucial

As suggested, we modified the text.

Line 93 This allows the standardisation of the evaluation procedures

As suggested, we modified the text.

Line 156 allows the identification of both

As suggested, we modified the text.

Line 162 In detail

As suggested, we modified the text.

Line 163 indeed prior selection manual training is needed.

As suggested, we modified the text.

Line 164 beginning the operator must choose samples relating to the

As suggested, we modified the text.

Line 168 and sets up

As suggested, we modified the text.

Better identifies

As suggested, we modified the text.

Line 170 the programme can be saved

As suggested, we modified the text.

Line 169 Are you sure n. of pixels makes sense? What is n?

We replaced “n.” with “number of pixels”. In this way it makes sense.

Line 172-173 max in full?

Yes, max in full.

Line 173 Selection amount refers to cereals

As suggested, we modified the text.

Line 218 The VIS absorptions yielding

As suggested, we modified the text.

Line 219 the Cucurachi

As suggested, we modified the text.

Line 224 was set at 80

As suggested, we modified the text.

Line 228 a solution (remove was prepared before a solution)

As suggested, we modified the text.

Line 229                        1 g of

As suggested, we modified the text.

Line 237 Place The phenolic compounds in a new paragraph

As suggested, we modified the text.

Line 177 With respect to the general settings, the Reject choice was set as follows, ie

As suggested, we modified the text.

Line 179 in front and behind, were activated

As suggested, we modified the text.

Line 180 On this screen it was possible to set the speed

As suggested, we modified the text.

Line 184 the solenoid valve remained

As suggested, we modified the text.

Line 188 valves need to the opened at once was set

As suggested, we modified the text.

Line 190 re-write lines 190 to 194

As suggested, we re-write lines 190 to 194.

Line 195 -198 re-write could not follow

Lines 195-198 mention the classes of olives considered, so it does not need to be re-written.

Figure 2 and text: Why does it refer to 2 choices when there are in fact 3? Clarify

The text and Figure 2 are consistent because 3 classes are mentioned in the text as well as in the figure. In fact, the text states this: “Two classes of olives were identified on the basis of ripeness (“green”, those with an early 196 stage of maturity, and “black”, those that are ripe). In addition, we also considered invaiate olives in the “mixed” class”.

Line 204 Delete on the other hand as far as milling was concerned.

As suggested, we modified the text.

Instead pls use : The mill used a stainless-steel Alfa

As suggested, we modified the text.

Line 206 Kneading took place for 40

As suggested, we modified the text.

Line 208 used a two-phase

As suggested, we modified the text.

Line 271 Are you sure the word is leave – do you refer to leaf or leaves?

As suggested, we replaced with “leaves”.

Line 276 A statistical ordination approach was

As suggested, we modified the text.

Line 276 not clear

As suggested, we re-written a sentence.

Line 278 olives. A principal component analysis (PCA) was conducted using the software

As suggested, we modified the text.

Table 2 preferably do not split the word olives

As suggested, we modified the text.

Line 301  obtained from the olives at different

As suggested, we modified the text.

Line 302 green olives were called EVOOmix

As suggested, we modified the text.

Line 303 olives were identified

As suggested, we modified the text.

Line 311  if you refer to a range should you not show highest and lowest figures?

Line 312 same argument as above

Ranges extreme values were highlighted in the table as suggested.

Line 317 400 and 500 nm, that correlated to the concentrations of carotenoids and between 500 and 700 nm that corresponded to

As suggested, we modified the text.

Line 323 Did not follow, possibly re-write

We replaced a word in order to make the meaning of the sentence clear.

Line 333 for the oils obtained from more ripe olives, EVOOSC showed a lower

As suggested, we modified the text.

Line 348 the lowest concentration

As suggested, we modified the text.

Line 349 where it was demonstrated

As suggested, we modified the text.

Line 355 The study showed

As suggested, we modified the text.

Line 357 error rate depended

As suggested, we modified the text.

Line 362 Therefore, the study

As suggested, we modified the text.

Line 370 After the application of the pattern recognition technique, an unsupervised Principal Component Technique (PCA) was applied to the……

As suggested, we modified the text.

Line 372 A biplot of the final PCA was then obtained.

As suggested, we modified the text.

Figure 4 legend Biplot of the PCA conducted on some of the chemical

As suggested, we modified the text.

Line 380 Along the first components

As suggested, we modified the text.

Line 382 – 385 please clarify sentence PESCARA

That was done, thanks.

Line 390 and optoelectronic ones in particular represent important tools implemented

As suggested, we modified the text.

Line 393 of the assessments, processing speed, capacity to work, (potentially) without interruption etc.

As suggested, we modified the text.

Line 396 to eliminate defective olives

As suggested, we modified the text.

Line 397 the potential of these technologies for use in EVOO production.

As suggested, we modified the text.

Line 398 The oil analyses obtained showed significant differences both chemically

As suggested, we modified the text.

Line 401  while still being able

As suggested, we modified the text.

Line 403 when both unripened and completely ripe

As suggested, we modified the text.

Line 407 thus are commonly concurrently harvested

As suggested, we modified the text.

Line 408 A selection potentially gives the opportunity

As suggested, we modified the text.

Line 411 Future work will test and evaluate

As suggested, we modified the text.

Round 2

Reviewer 1 Report

The authors have incorporated the required changes to the paper.

The paper is much clearer now.